# Halved Incidence of Scrub Typhus after Travel Restrictions to Confine a Surge of COVID-19 in Taiwan

**DOI:** 10.3390/pathogens10111386

**Published:** 2021-10-27

**Authors:** En-Cheng Lin, Hung-Pin Tu, Chien-Hui Hong

**Affiliations:** 1Department of Dermatology, Kaohsiung Veterans General Hospital, Kaohsiung 813, Taiwan; james820720@gmail.com or; 2Department of Public Health and Environmental Medicine, School of Medicine, College of Medicine, Kaohsiung Medical University, Kaohsiung 807, Taiwan; p915013@kmu.edu.tw; 3Department of Dermatology, School of Medicine, National Yang Ming Chiao Tung University, Taipei 112, Taiwan

**Keywords:** scrub typhus, rickettsial disease, *Orientia tsutsugamushi*, COVID-19

## Abstract

Scrub typhus is a rickettsial disease that is usually transmitted by mite exposure. Infected patients may present with a fever, fatigue, headache, and muscle pain. A blackish skin lesion, called eschar, is pathognomic. The mortality rate in untreated cases is high. The first case of scrub typhus in Taiwan was reported in 1908 during the Japanese colonization. In this article, using the National Infectious Disease Statistics System (NIDSS) from the Taiwan CDC, we analyzed the dynamic incidence of scrub typhus from 2016 to 2021, both seasonally and geographically. In addition, we asked whether the recent travel restrictions and social distancing policy in Taiwan (19 May to 27 July 2021), implemented due to the COVID-19 outbreak, would change the incidence of scrub typhus. The results showed that scrub typhus was most common in summer, with an incidence almost twofold greater than that in winter or spring. Most cases were identified in rural regions. Interestingly, there was a significant 52% reduction in the summer incidence in 2021, compared to the average summer incidence of the past 5 years. This reduction coincided with the countrywide lockdown measures and travel restrictions. The restricted measures for outdoor activities may have contributed to the reduced incidence of scrub typhus.

## 1. Introduction

Scrub typhus results from *Rickettsia* infection. Its clinical manifestation includes a fever, headache, chills, arthralgias, and myalgias. A specific cutaneous lesion, eschar, and centrifugal maculopapular rash become recognized as the disease evolves [1]. *Orientia tsutsugamushi*, belonging to the family Rickettsiaceae, is the pathogen responsible for scrub typhus. This bacterium was first reported by Hakuju Hashimoto in Japan in 1810 [2]. *Orientia tsutsugamushi* naturally survives in mite populations by transmission from females to eggs, eggs to larvae, and then to adults. The mite larvae, called chiggers, are natural ectoparasites of rodents [3]. When infected chiggers bite humans, the bacteria are transferred. 

The geographical distribution of scrub typhus has been described with the term “Tsutsugamushi Triangle”, an area ranging from Pakistan to the northwest, Japan to the northeast and northern Australia to the south [4,5]. However, the geographical distribution of scrub typhus is now recognized as being wider than the traditional Tsutsugamushi Triangle, because cases have now been reported in Africa and South America. Scrub typhus is currently transmitted mostly by mites that feed on rodents and live in scrubland. Is is there that they might also bite humans who happen to walk through the scrubland. This mite-borne scrub typhus is not directly transmitted from human to human. 

Depending on its seasonality, scrub typhus is categorized into summer types and winter types [6]. The summer type has a seasonal distribution, between March and November, with a peak in the summer between June and August. Conversely, the winter type has a peak occurrence in October. The summer type is prevalent in southern China; however, the autumn–winter type is prevalent in northern China. The key vectors are also different in these two types: the summer type is *Leptotrombidium deliense*, and the autumn–winter type is *L.*
*scutellare*. As a result, seasonal variations in scrub typhus infections may result from seasonal fluctuations of the larval chigger mites and their rodent hosts. In fact, scrub typhus in Taiwan is predominantly caused by the summer type. People should pay more attention to this illness during this season.

Taiwan, located between Southeast Asia and East Asia, lies in the center of the Tsutsugamushi Triangle. Tracing back to the history of scrub typhus in Taiwan, a case in Hualien was suspected by Japanese police officers in 1908 during the Japanese colonization [7,8,9]. In 1915, Juro Hatori made a connection between various reports of an unknown fever to tsutsugamushi disease and the role of red larval mites (*Leptotrombidium* spp.) in disease transmission [10]. In 1932, Naritomi characterized several scrub typhus cases in the Pescadores Islands, the outlying islands in the Taiwan Strait [11]. After the Korean war, the U.S. Naval Medical Research Unit No.2 (NAMRU-2) served as the regional headquarters throughout the Vietnam War, recognizing a considerable number of cases of scrub typhus among military personnel in the Pescadores Islands [12,13,14,15]. In 1967, Lien et al. reported that *L. deliense* was the vector for scrub typhus in the Pescadores Islands [16]. In 1970, Gale et al. harvested *O. tsutsugamushi* from free-living *L. deliense* and wild rats (*Rattus* species) which were captured near an outbreak among military personnel in eastern Taiwan [17]. Dr. Hsin-Chun Lee reported that the annual number of cases had increased from 39 in 1990 to 302 in 1999 [18]. From 2001 to 2004, the cases significantly increased in eastern Taiwan [19]. From 2004 to 2016, an average of 435 cases countrywide were confirmed annually [20]. Undoubtedly, scrub typhus has remained a significant infectious disease in recent years. In this article, we focus on the epidemiologic trend of scrub typhus in Taiwan, both seasonally and geographically, in the past five years.

Recent lockdown measures to prevent COVID-19 transmission significantly contained the transmission of the virus. For example, the social distancing policy for COVID-19 significantly reduced viral respiratory tract infections in children in Finland [21]. Steffen et al. showed that the annual case numbers of mosquito-borne diseases, including chikungunya, dengue, and malaria, were significantly reduced by approximately 90% due to travel restrictions and lockdowns in Switzerland [22]. However, in 2020, there has been an annual increase in case numbers of tick-borne encephalitis from 112 to 210. The author speculated that this paradoxical increase may have resulted from many more people engaging in activities such as outdoor hiking or playing sports during the lockdown [22]. Therefore, we are interested in whether the travel restrictions and lockdown measures for COVID-19 were able to reduce the incidence of scrub typhus, a prototypic tick-borne disease, which is more common in rural regions than it is in urban regions. 

Taiwan enjoyed a period of low COVID-19 case numbers for more than 1 year since the worldwide outbreak, until a recent surge in indigenous COVID-19 cases in mid-May, 2021. The Central Epidemic Command Center (CECC) in Taiwan issued a nationwide Level 3 COVID-19 epidemic warning from May 19 to July 27 and urged people to abide by and cooperate with Level 3 epidemic prevention and control measures, which included traffic restrictions and the closure of mountain hiking cabins in the national parks. Thus, it provided a window in which to assess whether the travel restrictions and social distancing policy to combat COVID-19 would increase or decrease the frequency of scrub typhus. 

## 2. Results

### 2.1. Scrub Typhus Is More Common in Summer than in Other Seasons 

We first asked in which season scrub typhus was the most common. All confirmed that cases were divided into four groups, based on their initial months of reporting (Figure 1). From 2016 to 2020, most cases were in summer (average annual cases at 153.6), followed by autumn (average annual cases 121.6), compared to average annual cases of 79 in both spring and in winter (Figure 1A). In fact, the number of cases was highest for every summer, except in 2016, when the case numbers in autumn were slightly greater than those in summer. Similarly, the incidence rate was the highest for every summer, followed by the autumn (Figure 1A). The increased incidence in the autumn of 2016 correlated with a peak in the autumn temperature in the years between 2016 and 2020 (Figure 1B).

### 2.2. Geographic Distribution of Scrub Typhus in Taiwan from 2016 to 2020

We then assessed whether geographic distribution would affect the incidence of scrub typhus. We categorized all confirmed cases of scrub typhus from 2016 to 2020 into seven groups, based on their geographical distributions (Figure 2). Furthermore, all these regions were grouped into two groups, based on urbanism, as defined by the Taiwanese government (Figure 3). In the past five years, the highest incidence was found in the outlying islands (up to 60 events per 100,000 person-years). The second most common location was in the eastern mountainous regions of Taiwan (roughly 20–25 events per 100,000 person-year). These two regions contributed the majority of reported cases of scrub typhus. The third most common location was in the “Kao–Ping” region, which represented the most tropical and southernmost area of Taiwan. For the rural and urban model, the general population in the urban regions was larger than that in the rural regions (approximately 17,000,000 and 6,000,000 persons, respectively). The incidence in the rural regions was five times higher than that in the urban regions in each year from 2016 to 2020. The difference in the incidence rate ratio between the urban regions and the rural regions exhibited quite a high statistical significance (Figure 3, *p* < 0.0001). 

### 2.3. A dramatic Decline in the Incidence of Scrub Typhus in Summer, 2021, Compared to Summers of the Previous Five Years, under Lockdown Measures against the COVID-19 Outbreak in Taiwan

COVID-19 has been classified as a pandemic since December 2019. COVID-19 rapidly spread globally from 2020 to May 2021, when the transmission of the disease gradually decreased. However, Taiwan enjoyed a period of low COVID-19 case numbers for more than 1 year since the worldwide outbreak, until a recent surge in indigenous COVID-19 cases in mid-May, 2021. Multiple strict lockdown policies from the Taiwan Centers for Disease Control (CDC) were applied, including social distancing, travel restrictions, and the closure of mountain hiking cabins in the national parks. Surprisingly, the average summer incidence during the past 5 years (0.65 per 100,000 person-years) decreased significantly in 2021 by 53.6% (to 0.3 per 100,000 person-years) (Figure 4). 

To investigate whether urbanism also affected the summer incidence, we also analyzed the distribution and incidence of scrub typhus from June to August 2021 and compared these with averages of the past five years (Figure 5). The results showed a significant decline in summer incidence in the regions of Taipei, Central, and Kao–Ping (Table 1). In the eastern region and outlying islands, where the majority of cases were reported, the summer incidence declined, although it did not reach a statistical significance. Nevertheless, the summer incidence declined in both the urban and rural regions with relatively high statistical significances.

## 3. Discussion

In this study, scrub typhus was most prevalent in summer, followed by autumn. Moreover, the key vector of scrub typhus in Taiwan was *L**. deliense*. All these characteristics are compatible with the summer type of scrub typhus. Due to the association between the high incidence and the high temperature in summer, it is possible that the high temperature might be related to the increased incidence in summer; however, the causality is difficult to establish. Nevertheless, the temperature data in the past five years correlated well with the incidence. In fact, autumn 2016 was exceptionally warm, leading to the highest incidence rate in that autumn among the years 2016–2020. However, summer is the rainy season, when typhoons occur in all regions of Taiwan. It remains uncertain as to whether humidity or windy and rainy typhoons would accelerate the transmission of scrub typhus or not.

Regarding the geographic distribution of scrub typhus in Taiwan, most cases were diagnosed in the Eastern region in the past five years. Similar to Nicholas T. Minahan et al.’s review [20], the annual incidence of scrub typhus, from 2016 to 2020, was the highest in the eastern regions; however, the stratification of the capital city, Taipei, and the southernmost Kaohsiung–Pingtung (Kao–Ping) regions revealed a much higher incidence in the outlying islands. Due to differences in the economic development, the eastern regions and the outlying islands are all rural regions, whereas all of the urban regions (three metropolitan regions: Taipei, Taichung, and Kaohsiung) are located in the west of Taiwan. Our results indicated that the incidence in the rural regions was significantly higher than that in the urban regions (*p* < 0.0001). Our data are consistent with the reports from Olson et al., who demonstrated that socioeconomic factors (increased urbanization and increased school enrollment) were associated with the decreased incidence of scrub typhus in 1979 [23]. In fact, most of these rural regions, as well as the eastern regions and outlying islands are the popular travel destinations. The travel activity may increase the summer incidence in these two regions. The reduced travel activities undoubtedly affects the incidence in urban regions. The incidence of scrub typhus was highest in the outlying islands and eastern regions, followed by Kao–Ping, which is the southernmost part of Taiwan. In fact, Taiwan is crossed by the Tropic of Cancer, which divides it into tropical (south) and subtropical (north) zones. In summer, the mean temperature of all regions can reach more than 30 °C. However, in winter, the mean temperature in the north of Taiwan can decrease to around 15 °C, but in the south of Taiwan only decreases to around 20 °C [24]. These geographic features might explain the relatively higher incidence of scrub typhus in the Kao–Ping (southern Taiwan) regions, followed by the eastern regions and the outlying islands.

COVID-19 was classified as a global pandemic in December 2019. Luckily, there was no widespread infection in Taiwan in 2020. However, hundreds of daily COVID-19 cases have been identified since mid-May 2021. The government exercised rigorous policies to control the viral spread. Traveling and crowd gathering are forbidden. Interestingly, there was a significant decrease in the incidence of scrub typhus in the summer 2021, compared with each summer from 2016 to 2020. The policies of travel restriction and social distancing may reduce the risk of coming into contact with rodents or other vector species. However, the reduction in incidence was not significant in the eastern regions or the outlying islands. The lockdown measures do not differ by region, as per the national order of the CECC. This implicates that if the travel restrictions during COVID-19 are not rigid enough to prevent all intercounty travel, travelers from urban regions may decide to travel to the eastern regions and outlying islands for leisure. This could result in the reduction in the overall disease incidence; however, the reduction might not be significant in the eastern region and outlying islands because both are common travel destinations and places where the most typhus cases are reported. We made an inference that the effect of lockdown measures was absent in those regions where scrub typhus was endemic. Indeed, it was travel restrictions, not social distancing, that reduced the incidence of the vector-borne disease. Local people, from those regions where scrub typhus was endemic, were still at risk due to engaging in outdoor activities, such as agricultural work. In summer 2021, there was actually no significant decrease in the incidence in the eastern regions and the outlying islands. In the outlying islands, the incidence was not even the lowest of all the years analyzed. Nevertheless, a significant decline in incidence was demonstrated in both the urban and nonurban regions. The pandemic prevention policies restricted people’s activity, thus preventing the risk of exposure to mites or other vectors. In other words, staying at home not only prevented people from contracting COVID-19, but also from scrub typhus.

There are still some limitations to our study. First, the effects of vegetation in different regions may interfere with the transmission of scrub typhus. However, we do not have authority to access the national vegetation data. Second, the patient’s socioeconomic status could not be evaluated in this study due to data unavailability based on NIDSS. Third, the data was collected from NIDSS in early September. There might be few cases at the end of August whose reporting to NIDSS was delayed. The number of cases in summer 2021 might be underestimated. However, when the data was accessed again on Oct 20 during the resubmission, the case numbers in this summer did not change.

## 4. Methods

All demographic data of scrub typhus, including new reported cases per month and per county from 2016 to 2021, were acquired from a public domain from the Taiwan National Infectious Disease Statistics System (NIDSS), Taiwan CDC (https://nidss.cdc.gov.tw/) (accessed on 20 October 2021) [25]. The reported cases are based on the symptomatic date rather than the date of diagnosis or the date of tick bite. Based on the NIDSS statistical measurements, the regions were also classified based on administration and geography, to seven groups, which included Taipei, Northern, Central, Southern, Kao–Pin, Eastern, and Outlying Islands, including the Pescadores, Quemoy, Matsu Islands, etc. For reporting to NIDSS, patients’ residential regions were used rather than the regions where they were infected. 

For estimation of the dynamic incidence of scrub typhus, general population statistics (including the numbers of general population each month in each county) from 2016 to 2021 were obtained from the Taiwan Statistics Bureau [26] to estimate the country- and county-specific disease incidence. The meteorological seasonal system was utilized to define the months in each season in our study. Meteorological seasons were classified by temperature, with summer being the hottest quarter of the year and winter the coldest quarter of the year. The seasons were classified based on the average monthly temperature from the Taiwan Central Weather Bureau [24]. Hence, in Taiwan, spring starts on March 1st, summer on June 1st, autumn on September 1st, and winter on December 1st. The urban or rural regions were categorized based on the administrative districts defined by the Taiwan government. The urban regions refer to metropolitan cities or towns with a population of more than 300,000. Those conurbations were home to populations smaller than those in rural regions. Monthly temperature statistics during the past 5 years were obtained from the Central Weather Bureau of Taiwan (http://www.cwb.gov.tw/V7/climate/climate_info/statistics/statistics_1_3.html) (accessed on 20 October 2021). 

For statistical analysis, the incidence rate or incidence rate ratio (IRR) was calculated using a generalized linear model to perform Poisson regression analysis, which is a log-linear model. A *p*-value of less than 0.05 (*p* ≤ 0.05) was deemed to be statistically significant. Microsoft Excel 2016 and SAS statistical package version 9.4 (SAS Institutes, Cary, NC, USA) were used for statistical analysis.

## 5. Conclusions

Scrub typhus is more common in summer than in the other seasons in Taiwan. From 2016 to 2020, most cases occurred in the eastern regions and outlying islands. The incidence was more common in rural regions than in urban regions. In mid-2021, the number of scrub typhus cases decreased by more than 50% after travel restrictions and social distancing policies were imposed to fight against the COVID-19 outbreak.

## Figures and Tables

**Figure 1 pathogens-10-01386-f001:**
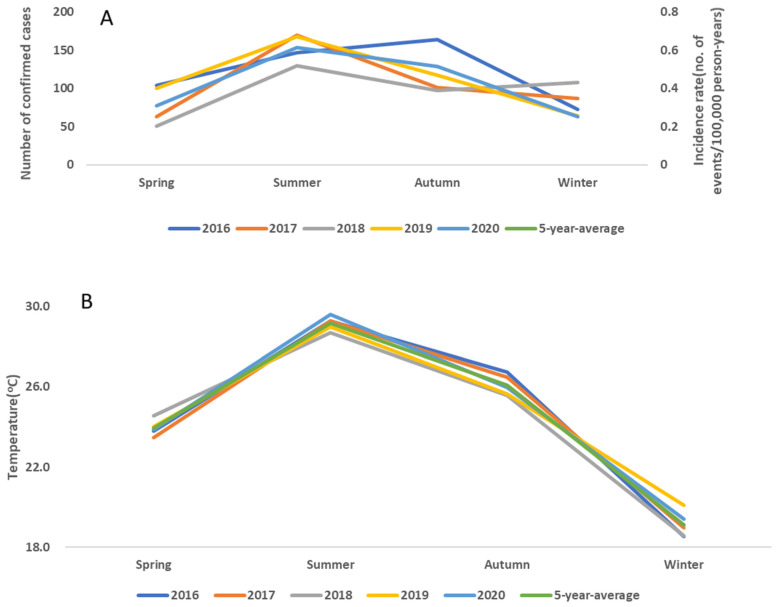
All scrub typhus cases from 2016 to 2020 were divided into four groups (Spring: March/April/May; Summer: June/July/August; Autumn: September/October/November; Winter: December/January/February). The number of new cases and the incidence (**A**) were both highest in the summers of the past 5 years, except in 2016, when case numbers peaked in autumn. The incidence was measured by season and by year. The incidence was the highest in each summer. (**B**) The autumn temperature during the past 5 years peaked in 2016.

**Figure 2 pathogens-10-01386-f002:**
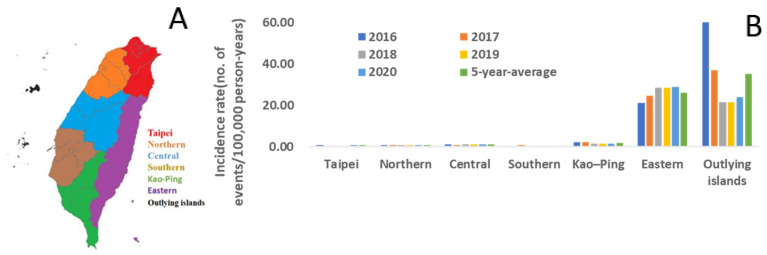
(**A**) All confirmed cases from 2016 to 2020 were categorized into seven groups, based on the regions where the patients lived. (**B**) Region-specific incidence was measured in each year. The majority of cases were located in the Outlying Islands and the Eastern region of Taiwan.

**Figure 3 pathogens-10-01386-f003:**
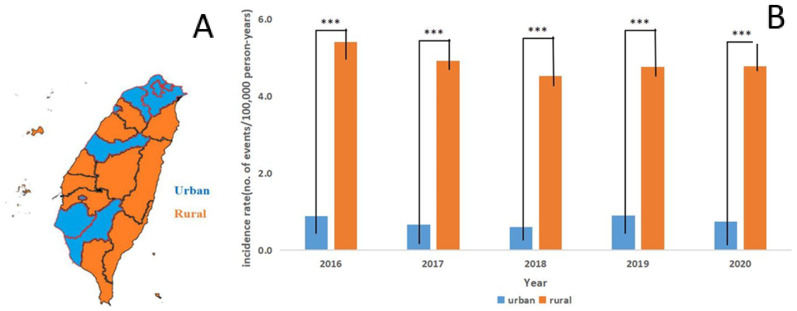
(**A**) Scrub typhus cases were grouped into urban or rural regions, based on the administrative districts defined by the Taiwan government. (**B**) The incidence in the rural regions was significantly higher than that in the urban regions in each year from 2016 to 2020. *** *p* < 0.0001.

**Figure 4 pathogens-10-01386-f004:**
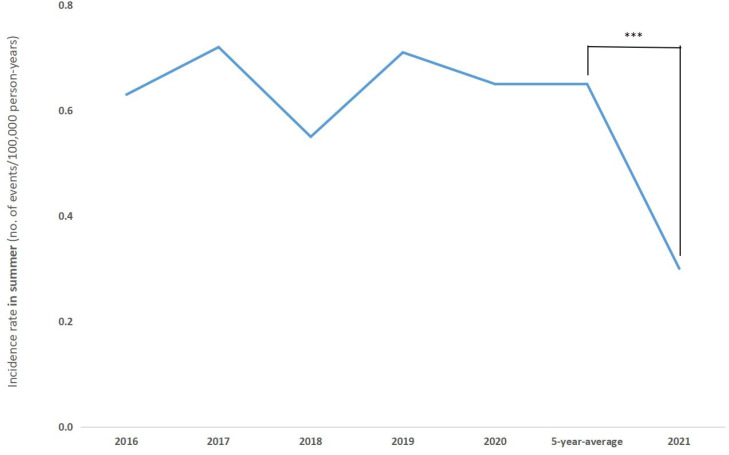
Incidence of scrub typhus in summer from 2016 to 2021 were measured. In summer 2021, the number of confirmed cases sharply decreased by more than 50%, as compared with those in the previous 5 years. *** *p* < 0.0001.

**Figure 5 pathogens-10-01386-f005:**
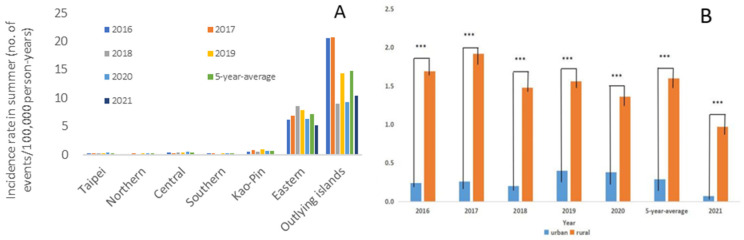
The incidence in summer was categorized the by geographical region (**A**) and urbanism (**B**). The Outlying Islands had the highest incidence rate, both in summer 2021 and in the 5-year summer average. In addition, in each summer, the incidence rate in the rural regions was significantly higher than that in the urban regions. *** *p* < 0.0001.

**Table 1 pathogens-10-01386-t001:** The incidence rate ratio (IRR) in summer between 2021 and the 5-year average was applied in different regions and urban areas. The incidence rate ratio (IRR) was calculated using the generalized linear model.

	Incidence Rate Ratio	
Region	In 2021 vs. 5-Year Average	*p* Value
All	0.48 (0.36–0.63)	<0.0001
Taipei	0.22 (0.08–0.66)	0.0069
Northern	0.54 (0.16–1.84)	0.3274
Central	0.29 (0.11–0.8)	0.0162
Southern *	-	-
Kao-Pin	0.12 (0.04–0.41)	0.0006
Eastern	0.73 (0.45–1.18)	0.1995
Outlying islands	0.71 (0.43–1.16)	0.1717
Urban	0.23 (0.13–0.44)	<0.0001
Rural	0.60 (0.44–0.84)	0.0023

* The incidence rate ratio was not calculated, due to the incidence rate per 100,000 person-years being 0.0 in 2021 and 0.2 in the 5-year average.

## Data Availability

Data are available from Taiwan Centers for Diseases Control and Department of Household Registration, Ministry of the Interior, Taiwan.

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
