# Peer review of "Halved Incidence of Scrub Typhus after Travel Restrictions to Confine a Surge of COVID-19 in Taiwan"

_pathogens, 2021, doi:10.3390/pathogens10111386_

Round 1
Reviewer 1 Report
Overall, it's a well-written article. It's not as original as it stands out, but it seems to represent the phenomenon well with data.
Author Response
Dear reviewer,
We have revised the manuscript. Thanks for your advice and suggestion.
Best regards. Dr. En-Cheng, Lin E-mail: james820720@gmail.com

Reviewer 2 Report
The effect of the COVID-19 pandemic on other diseases is an interesting one, so this submission is of relevance. However, it requires major revision, particularly around the methodology and presentation of results. Specifically, expert epidemiological and statistical assistance in the data analysis is required.
The title can be shortened.
English requires improvement, especially in the abstract.
Geographical distribution of ST is now recognised as being larger than the traditional T-Triangle. Evidence for the disease has now been found in Africa and South America.
Presentation of data: ST is an acute disease, so the term incidence is more appropriate than prevalence. The incidence rates are more important than the numbers of cases, for basic epidemiological reasons. The source and derivation of population denominators used in the calculation of the incidence rates should be more clearly explained in the methods. Likewise the rules used for classification of urban and rural areas. These terms should be used throughout, rather than ‘city’ and ‘county’.
The number of graphs and tables can be substantially reduced. Figure 1 and the following two un-numbered graphs, the un-numbered graph following Figure 2, and maybe Figure 4, can all be dispensed with. Figures 2 and 3 can be merged with appropriate graphics used to demonstrate the map information.
In terms of the purpose of the submission, the most important variable is the summer incidence rate, and how the rate for 2021 differs from the average of previous years, and whether this is statistically significant – both overall, and between the regions of Taiwan. (Figure 5, but with incidence rates rather than numbers). No more than one or two decimal places is required for the rates. The appropriate statistical analysis should be applied in consultation with a statistician. The more general issues of seasonality and urban vs rural can be covered in the discussion.
Author Response
Dear reviewer,
Thanks for your advice and recommendation.
- The title has been shortened.
- Incidence was appropriately calculated. The incidence rate or incidence rate ratio (IRR) was calculated using the generalized linear model to perform Poisson regression analysis, which is a log-linear model.
- We also adjusted the number of graphs and tables.
We appreciated your advice again.
Best regards. Dr. En-Cheng, Lin E-mail: james820720@gmail.com

Reviewer 3 Report
Generally speaking, I applaud the authors for looking for impacts of the COVID pandemic others than COVID cases themselves. But the paper needs much improvement.
Introduction:
Describe how the mites are transmitted! In Europe typhus was once transmitted by lice. Lice were transferred from human to human for example by sharing clothing or bedding or by close contact, usually in poor sanitary conditions. Scrub typhus is transmitted by mites that feed on rodents and live in scrubland. There they might also bite humans that happen to walk through the scrubland. As far as I know they are not transmitted from humans to humans directly. This has an important impact on which measures will affect transmission rates and how. Social distancing will have no big effect. Reducing travel and reducing walks in the wilderness will have. As the authors point out themselves, lockdown measures in Switzerland increased the rate of tick-borne encephalitis. Transmission route is similar for mite-borne diseases.
The different types of scrub typhus (summer and winter type) are only explained in the discussion. Maybe this part would be fit for the introduction as well. Readers from other parts of the world are not familiar with that disease. It would be good to provide full information on that disease in the introduction.
Methods:
This section is much too short! Describe the data! The “demographic data” are in fact “health data” or “incidence data”. I am not sure which information is provided with the reporting of the disease incidence: Apparently you use information on time (month or higher resolution?) and on place (region)? Is it the date of diagnosis or the date of the bite? Is it the place of living or place of infection? Especially if you are interested in the effects of travel restrictions this is an important issue. Urban dwellers might have used to visit scrubland for vacation purposes. With travel restrictions they would no longer be able to do so. That would have an effect on the disease rate of urban dwellers, even if the infection occurs in rural areas.
Define the seasons! Depending on the regional climate seasons have different meanings. I assume (but you never explain) that by e.g. “spring” you mean the months March-May. And describe the meteorological characteristics per season and per region. Only with this knowledge can the reader appreciate your findings!
Results:
Perform some analytical statistics! Influence of temperature? Reasons for seasonality? There might be a reason why in 2016 the highest rates were observed in autumn! Maybe summer 2016 was exceptionally cold or rainy? Or the autumn was exceptionally warm?
Report regional differences relative to population number, not absolute numbers! You report “prevalences” also in figure 2. These relative numbers are much more meaningful than the absolute numbers in the top part of the figure. But I am not sure if these are prevalences really and not incidences!
Differences in lock-down measures by region? Regression analysis: numbers per district (?) in 2021 are associated with numbers in previous years + lock-down measures + temperature (difference). Urbanity, socioeconomic status, vegetation? Only in the discussion do you hint at differences per region (line 179ff). This claim needs substantiation by a statistical analysis reported in the results.
Author Response
Dear reviewer,
Thanks for your advice and recommendation.
- method: Incidence was appropriately calculated. The incidence rate or incidence rate ratio (IRR) was calculated using the generalized linear model to perform Poisson regression analysis, which is a log-linear model.
We also explained the meteorological seasonal system which was used in our study.
2. result: we added incidence rate and incidence rate ratio in this part. A p-value less than 0.05 (p ≤ 0.05) is statistically significant.
We are so grateful for your advice and suggestions.
Best regards,
Dr. En-Cheng, Lin E-mail: james820720@gmail.com

Round 2
Reviewer 2 Report
The revised manuscript is substantially better than the original submission. The mention of louse-borne typhus is irrelevant and should be removed. The first figure can be improved, and there are some grammatical and typographical corrections required - see annotated comments on the uploaded PDF version.

Author Response
Comments and Suggestions for Authors
The revised manuscript is substantially better than the original submission. The mention of louse-borne typhus is irrelevant and should be removed. The first figure can be improved, and there are some grammatical and typographical corrections required - see annotated comments on the uploaded PDF version.
Response:
Thanks again for these comments on our manuscript. The mention of louse-borne typhus is removed. We also merged Figure 1A and Figure 1B per your suggestion. All the term of ”urbanity” is changed to “urbanism”. All scientific genus and species names are italicized.
Best regards,
Chien-Hui Hong, MD, PhD
Department of Dermatology, Kaohsiung Veterans General Hospital, Kaohsiung 807, Taiwan
Email: zieben@gmail.com
Reviewer 3 Report
The paper has been improved much and the authors have responded to my concerns.
It was not necessary to add my example regarding typhus in Europe. That was a problem in war times in the 19th century, but is currently not an issue. I meant that example to show that rickettsia can be transferred by close contact between humans (in poor sanitary conditions). But with scrub typhus this seems to be not the case. Therefore, social distancing per se will not have an effect on the transmission. The main aim of the COVID-19 measures was to reduce social contacts. The authors for example report the experience from Switzerland, where some diseases that were usually acquired in foreign countries like chikungunya, dengue, and malaria, were reduced, but endemic diseases like tick-borne encephalitis were increased. Indeed, it is travel restrictions, not social distancing, that reduced the incidence of vector-borne diseases.
Therefore, I would expect an effect of COVID-19 measures people in those regions where scrub typhus is not endemic. Where it is endemic, local people will still spend time outdoors (e.g. in agricultural work) and will still be at risk. And really, in the 2 regions with the highest scrub typhus incidence, there was no significant decrease in incidence in the summer 2021. In the outlying islands, the incidence was not even the lowest of all years analyzed. The reduction is seen in those regions where the disease is not endemic. I think it is worth pointing that out.
Summer is defined as June-August. The paper was written and submitted early in September. I wonder if there is some delay in reporting of part of the cases. In that case, number of cases in summer 2021 might be underestimated. If the authors have no information on reporting delays, they could maybe report this as a caveat?
Author Response
Comments and Suggestions for Authors
The paper has been improved much and the authors have responded to my concerns.
It was not necessary to add my example regarding typhus in Europe. That was a problem in war times in the 19th century, but is currently not an issue. I meant that example to show that rickettsia can be transferred by close contact between humans (in poor sanitary conditions). But with scrub typhus this seems to be not the case. Therefore, social distancing per se will not have an effect on the transmission. The main aim of the COVID-19 measures was to reduce social contacts. The authors for example report the experience from Switzerland, where some diseases that were usually acquired in foreign countries like chikungunya, dengue, and malaria, were reduced, but endemic diseases like tick-borne encephalitis were increased. Indeed, it is travel restrictions, not social distancing, that reduced the incidence of vector-borne diseases.
Response:
Thanks again for your generous advice. The mention of louse-borne typhus was already removed. Please refer to the “Introduction”.
Therefore, I would expect an effect of COVID-19 measures people in those regions where scrub typhus is not endemic. Where it is endemic, local people will still spend time outdoors (e.g. in agricultural work) and will still be at risk. And really, in the 2 regions with the highest scrub typhus incidence, there was no significant decrease in incidence in the summer 2021. In the outlying islands, the incidence was not even the lowest of all years analyzed. The reduction is seen in those regions where the disease is not endemic. I think it is worth pointing that out.
Response:
We completely agreed your view in that no decreased incidence in those regions where scrub typhus is endemic. We added this important viewpoint in the discussion. Please refer to Line 268-275.
“We made an inference that the effect of lockdown measures did not exist in those regions where scrub typhus was endemic. Indeed, it is travel restrictions, not social distancing, that reduced the incidence of the vector-borne disease. Local people, from those regions where scrub typhus was endemic, were still in risk due to engaging in outdoor activities, such as agricultural work. In summer 2021, there was actually no significant decrease in incidence in the Eastern region and Outlying Islands. In the outlying islands, the incidence was not even the lowest of all years analyzed.”
Summer is defined as June-August. The paper was written and submitted early in September. I wonder if there is some delay in reporting of part of the cases. In that case, number of cases in summer 2021 might be underestimated. If the authors have no information on reporting delays, they could maybe report this as a caveat?
Response:
This could be one of the limitations in our study. Please refer to Line 284-287.
“Third, the data was collected from NIDSS in early September. There might be few cases at the end of August delayed reporting to NIDSS. Number of cases in summer 2021 might be underestimated. However, when the data was accessed again on Oct 20 during the re-submission, the case numbers in this summer does not change.”
Best regards,
Chien-Hui Hong, MD, PhD
Department of Dermatology, Kaohsiung Veterans General Hospital, Kaohsiung 807, Taiwan
Email: zieben@gmail.com